# A molecular descriptor of intramolecular noncovalent interaction for regulating optoelectronic properties of organic semiconductors

Meihui Liu[1], Xiao Han[2], Hao Chen[2], Qian Peng [1] ✉ & Hui Huang [2] ✉

In recent years, intramolecular noncovalent interaction has become an important means to modulate the optoelectronic performances of organic/polymeric semiconductors. However, it lacks a deep understanding and a direct quantitative relationship among the molecular geometric structure, strength of noncovalent interaction, and optoelectronic properties in organic/polymeric semiconductors. Herein, upon systematical theoretical calculations on 56 molecules with and without noncovalent interactions (X···Y, X = O, S, Se, Te; Y = C, F, O, S, Cl), we reveal the essence of the interactions and the dependence of its strength on the molecular geometry. Importantly, a descriptor $S$ is established as a function of several basic geometric parameters to well characterize the noncovalent interaction energy, which exhibits a good inverse correlation with the reorganization energies of the photo-excited states or electron-pumped charged states in organic/polymeric semiconductors. In particular, the experimental $^1H$, $^{77}Se$, and $^{125}Te$ NMR, the optical absorption and emission spectra, and single crystal structures of eight compounds fully confirm the theoretical predictions. This work provides a simple descriptor to characterize the strength of noncovalent intramolecular interactions, which is significant for molecular design and property prediction.

Noncovalent interactions play important roles in chemistry, biology, catalysis, and material sciences[1–10]. Many experimental and theoretical works focus on intermolecular noncovalent interactions since they are directly responsible for the formation of molecular crystals, large clusters, folding of proteins, and binding of DNA and drugs[11–14]. However, intramolecular noncovalent interactions are less investigated, although they are also important to determine the structure of basic units for building larger aggregates[15–17]. It is noteworthy that the intramolecular interactions significantly influence the conformation of the organic/polymeric semiconductors, which is critical to determine their physicochemical properties, such as optical properties and

charge transport mobilities[18–20]. Many groups have adopted the noncovalent intramolecular interactions as an important strategy for designing high-performance organic/polymeric semiconductors for different applications, including organic solar cells (OSCs)[21–28], thin film transistors (OTFTs)[29–32], photodetectors (OPDs)[33] and light-emitting diodes (OLEDs)[34–37], since Huang, Marks, Facchetti, and coworkers termed this intramolecular noncovalent interaction as "Noncovalent Conformational Locks (NoCLs)" owing to their conformation-locking feature in enhancing the planarity and rigidity of organic semiconductors[38]. For example, the noncovalent fused ring electron acceptors (NFREAs) have been developed using the NoCLs

[1]School of Chemical Sciences, University of Chinese Academy of Sciences, Beijing 100049, P. R. China. [2]College of Materials Science and Opto-Electronic Technology & CAS Center for Excellence in Topological Quantum Computation & Key Laboratory of Vacuum Physics, University of Chinese Academy of Sciences, Beijing 100049, P. R. China. ✉e-mail: qianpeng@ucas.ac.cn; huihuang@ucas.ac.cn

strategy, which greatly improved the power conversion efficiencies (PCEs) to reach over 15%, similar to those of the fused ring electron acceptors (FREAs), while significantly decreased the synthetic complexity[25–28]. Moreover, various NoCLs (eg. S···O, Se···O) have been used to enhance the charge transport mobilities of organic/polymeric semiconductors for OTFTs[29], affording the record mobility as high as 14.9 cm² V⁻¹ s⁻¹[31]. Also, the OLED based on the thermally activated delayed fluorescence (TADF) obtained by the NoCLs strategy showed a high external quantum efficiency of 23.2%[39]. Although these investigations presented the efficacy of NoCLs in designing high-performance organic/polymeric semiconductors, they lack direct correlation between chemical structures, the strengths of NoCLs, and

the optoelectronic properties of organic/polymeric semiconductors. Furthermore, the nature of intramolecular NoCLs in organic/polymeric semiconductors, whether they arise from orbital overlap or electrostatic interactions, remains controversial[40–47].

Although many efforts have been made, it is very challenging to deeply investigate the fundamentals of NoCLs. First, noncovalent interactions are very weak forces, usually one to two orders of magnitude smaller than covalent interactions[48,49], which dramatically increased the difficulty of experimental characterization. Moreover, due to various chemical elements or groups, there is a wide variety of noncovalent intramolecular interaction, such as cation-π, anion-π, chalcogen-π, carbonyl-π, hydrogen bonding, halogen bonding, and

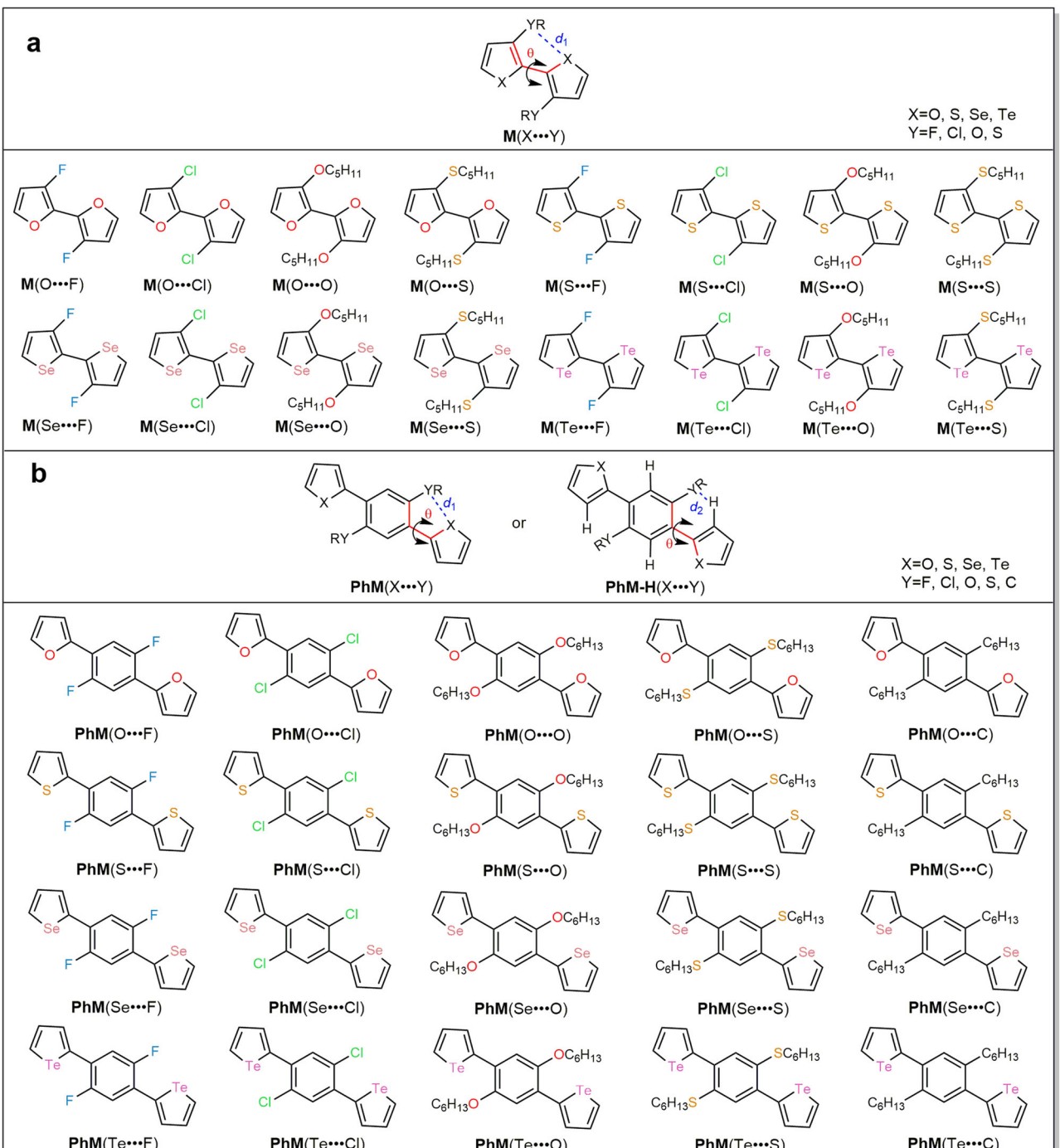

**Fig. 1 | Molecular structures of designed 56 compounds. a** Type I is denoted as **M**(X···Y) (16 molecules). **b** Type II is defined as **PhM**(X···Y) when θ is between 0° and 90° (20 molecules) and **PhM-H**(X···Y) when θ is between 90° and 180° (20 molecules in Figure S1 of Supplementary Information).

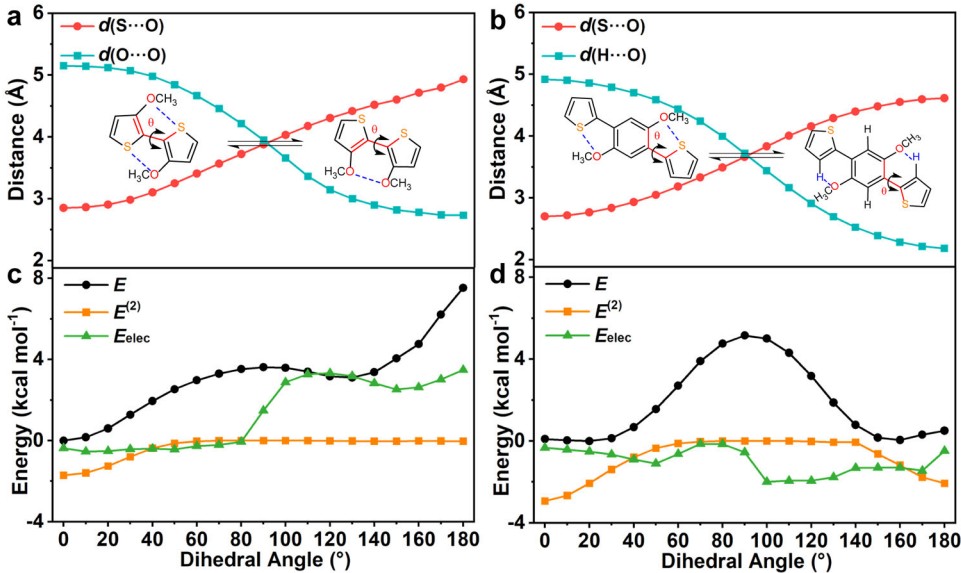

**Fig. 2 | The scan map as the dihedral angle changes.** The evolution of **a**, **b** the atomic distances between S and O atom (circle), and between O and O atom (square), and the conversion between two conformations given as insets, **c**, **d** the potential energy $E$, the variation of orbital energy $E^{(2)}$ and electrostatic interaction energy $E_{elec}$ as a function of the θ for **M**(S···O) and **PhM**(S···O).

chalcogen-bonding in organic molecules[50–53]. Thus, the theoretical study of these interactions became very complicated and challenging.

In this work, 56 molecules (Fig. 1) with various types of NoCLs (X···Y, X = O, S, Se, Te; Y = C, F, O, S, Cl) are used as models to systematically investigate the nature and strength of NoCLs by computational and experimental methods. It is revealed that the orbital interactions played dominant roles in the formation of NoCLs for these systems. Accordingly, a simple descriptor as a function of several molecular geometric parameters is established, which could characterize the strength of NoCLs without any complicated theoretical calculations. More importantly, the experimental [1]H, [77]Se, and [125]Te NMR, the optical absorption and emission spectra, and the single crystal structures of eight synthesized compounds fully confirmed the theoretical predictions.

## Result and discussion
### The nature of noncovalent interactions
The molecules **M**(S···O) and **PhM/PhM-H**(S···O) were selected as representatives to investigate the feature of X···Y NoCLs. The relaxed potential energy ($E$) surface (PES) scans were performed along the dihedral angle θ (marked in Fig. 1) from 0° to 180° for type I and type II. Based on the scanned geometry on the PES, natural bond orbital (NBO) analysis[54] was performed to obtain the variation of energy ($E^{(2)}$), and took atomic dipole moment-corrected Hirshfeld (ADCH) charge[55] (Figure S2) to calculate the electrostatic interaction energy ($E_{elec} = \frac{q_i q_j}{4\pi\varepsilon_0 \varepsilon d_{ij}}$), which both contributed to the total potential energy $E$. The results were plotted in Fig. 2, including $E$, $E^{(2)}$, and $E_{elec}$, as well as the distances between the atoms. From Fig. 2, it is easily seen that the equilibrium geometry of **M**(S···O) appeared at θ = 0.01°, whose $E$ was set to zero on the PES. At equilibrium geometry, the S···O distance $d$(S···O) of 2.85 Å is much shorter than the sum of van der Waals radii ($d$v) of S and O atoms $d$v(S···O) of 3.32 Å; the $E_{elec}$ is a small negative value due to few charges on S ($q_S$) and O ($q_O$) atoms (Figure S2); and the $E^{(2)}$ is a large negative value owing to large overlap between the n-orbital (lone pair electron of the oxygen atom) and the σ*-orbital of the S-C bond (n(O) → σ*(S-C)) (Figure S4). These indicate that the S···O NoCLs in **M**(S···O) are formed by the orbital interactions. Differently, the two minima on the PES of type II are very close in energy with a large energy barrier (Fig. 2d). Hence, the S···O interaction is examined

in **PhM**(S···O) with 0° ≤ θ ≤ 90°, and the H···O interaction is detected in **PhM-H**(S···O) with 90° < θ ≤ 180°. As shown in Fig. 2d, the orbital interactions are greatly stronger than the electrostatic interaction at the minima of **PhM**(S···O) (θ = 18.92° and $d$(S···O) = 2.76 Å), and both the orbital interactions and the electrostatic interactions are weak at the minimal of **PhM-H**(S···O) (θ = 157.34° and $d$(H···O) = 2.31 Å). The orbital interaction in **PhM**(S···O) comes from n(O)→σ*(S-C) and that in **PhM-H**(S···O) stems from n(O)→σ*(H-C).

The nature of NoCLs was examined at the equilibrium geometry of the studied systems. The equilibrium geometries were optimized and the frequencies were calculated at the B3LYP(D3)/6-31 + G(d) level[56] for the studied systems. For type II systems, the energy differences between two minima (ΔE = $E_{PhM}$ − $E_{PhM-H}$) were calculated to determine a more stable minimum, which would be investigated below (see Fig. 3a and Table S1). It is obviously found that **PhM-H**(O···Y) is more stable than **PhM**(O···Y), while **PhM**(X···O) (X = S and Se) and **PhM**(Te···Y) (Y = F, O, Cl, and S) are more stable than their corresponding **PhM-H**. The differences in energies between two conformations are small for the others. The potential energy surfaces of the systems were further scanned at the same level and plotted in Figure S3. Herein, three systems exist two conformations, **PhM**(X···C) and **PhM-H**(X···C) (X = S, Se, and Te), owing to a low energy barrier (<2 kT ≈ 1.2 kcal mol⁻¹) of mutual transformation between them, and the others have only one conformation due to large energy barrier.

We examined the distances, electrostatic interaction, and orbital interaction of the X···Y noncovalent bonds of the stable 36 molecules (16 **M**(X···Y) ones, 9 **PhM**(X···Y) ones, and 11 **PhM-H**(X···Y) ones). It is generally believed that the NoCLs are effectively formed between two atoms if the distance between them is less than $d$v. According to the Δ$d$ = $d$v-$d$ results given in Fig. 3b, 20 of the 36 compounds have significantly large Δ$d$ (>0.10 Å), being conducive to the formation of NoCL. The nature of NoCLs is either electronic interaction or orbital interaction[40,45]. From Fig. 3c, it is seen that the electrostatic attraction interactions of X···Y in the 36 molecules are very small, ranging from −2.21 to 0.00 kcal mol⁻¹, which is impossible to be responsible for the NoCLs. The orbital interaction varies greatly with $E^{(2)}$ ranging from −5.27 to 0.00 kcal mol⁻¹ (Fig. 3d) for the systems with and without NoCLs, which is likely to be the main contributor to form NoCLs. Of 36 compounds, 15 with $E^{(2)}$<−1.00 kcal mol⁻¹ are considered to have

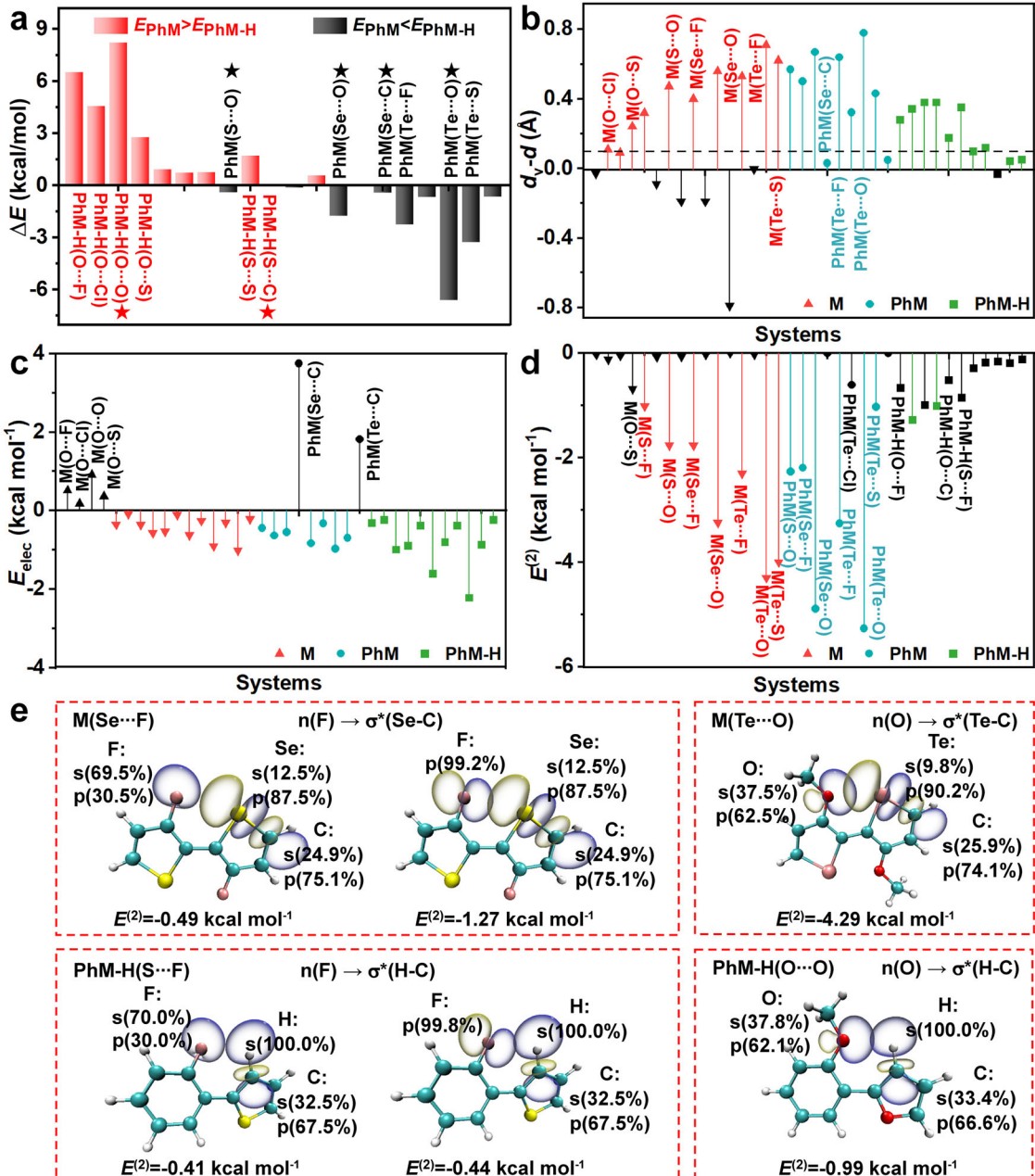

**Fig. 3 | Energy differences and noncovalent interactions obtained by various calculation methods. a** Energy differences between **PhM** and **PhM-H** ($\Delta E = E_{PhM} - E_{PhM-H}$). The red bar represents that **PhM-H** is more stable than **PhM**, and the black bar denotes that **PhM** is more stable than **PhM-H**. The ones marked with stars are experimentally synthesized. **b** The values of the sum of van der Waals radii of two atoms ($dv$) minus the distance between the two atoms ($d$), $\Delta d = dv-d$. **c** The electrostatic interaction energy $E_{elec}$. **d** The orbital interaction energy $E^{(2)}$, which is maked black as its absolute value <1.00 kcal mol$^{-1}$. **e** NBO overlap between the n-orbital and the σ*-orbital, in which the percentage represents the contribution of atomic orbitals participating in hybridization.

significant NoCLs, and the strength of NoCL exhibits the order of S⋯F < Se⋯F < Te⋯F, S⋯O < Se⋯O < Te⋯O, and X⋯F < X⋯O (X = S, Se, and Te). The orbital interaction is determined by the type, shape, and orientation of orbitals and bonds. As shown in Fig. 3e and S4, the interactions mainly happen between the n-orbital provided by lone pair electron on heteroatom Y in the side chain and the antibonding σ*-orbital of X-C or C-H bond in chalcogenide ring, n(Y)→σ*(X-C) or n(Y)→σ*(H-C). Considering the composition of the orbitals, there are different types of orbital interactions, such as n(s)→σ*(sp²), n(p)→σ*(sp²), and n(sp)→σ*(sp²) for **M** and **PhM** molecules, and n(s)→σ*(s), n(p)→σ*(s) and n(sp²)→σ*(s) for **PhM-H** molecules. The n(sp²)→σ*(sp²) shows the strongest interaction when p-orbital component dominates in the hybrid sp²-orbital, because the shape and

orientation of the sp²-orbital are favorable for overlap, such as Te⋯O NoCL. While the participation of s-orbital weakens the interaction due to its very small electronic density distribution area, like H⋯Y NoCL. Hence, the H⋯Y NoCLs in **PhM-H** molecules are always weaker than X⋯Y NoCLs in **M** and **PhM** molecules. Overall, 15 of 36 molecules were predicted to have significant NoCLs, namely, X⋯Y (X = S, Se, and Te, and Y = F and O) and Te⋯S in **M** and **PhM**, and H⋯S and H⋯Cl in **PhM-H** (Figure S1), which are mainly controlled by the orbital interactions.

**The establishment of the descriptor for noncovalent interaction**
It is very significant to establish the quantitative relationship among the molecular geometry, the strength of NoCL, and the photophysical property, which would directly guide the molecular design for organic

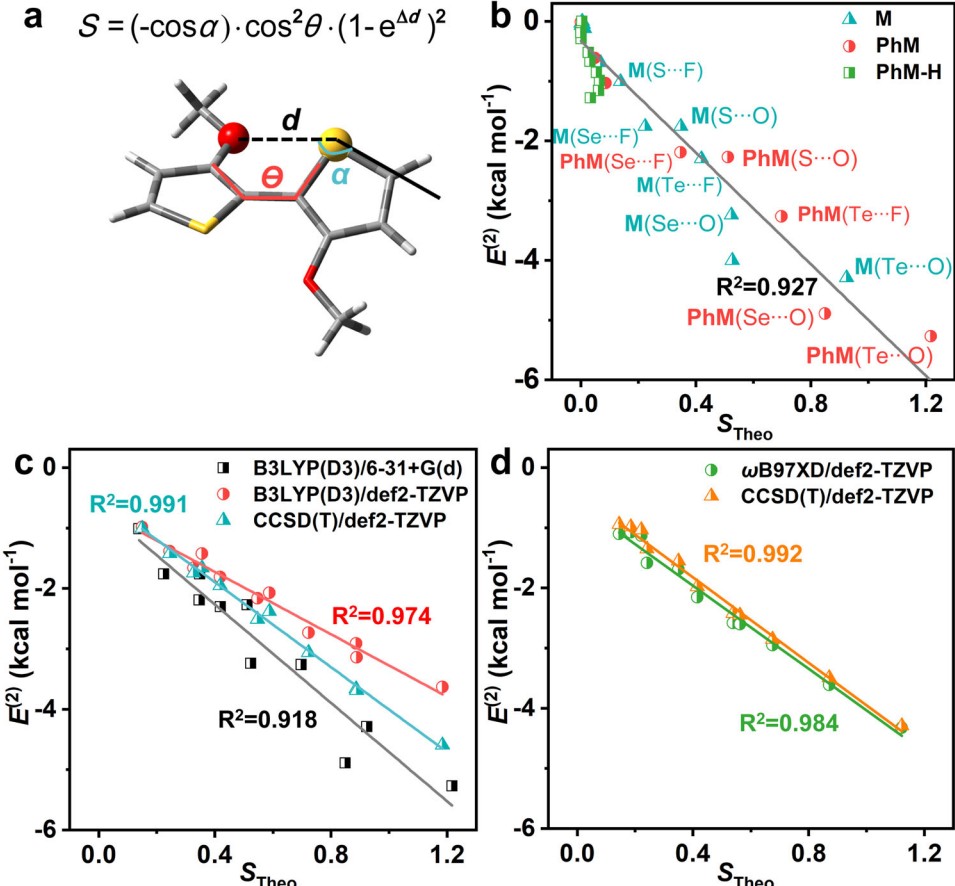

**Fig. 4 | The establishment of the descriptor for NoCLs. a** The geometrical parameters of the computational model. **b** The correlation between $E^{(2)}$ and $S$ with fitted line at B3LYP(D3)/6-31 G + (d) level for 36 compounds. **c, d** The $E^{(2)}$ versus $S$ at different theory levels based on the optimized geometries with fitted lines by B3LYP(D3)/def2-TZVP and ωB97XD/def2-TZVP, respectively, for 11 compounds with apparent noncovalent interactions.

optoelectronic materials. As it is known from the above discussions, the NoCLs in the studied systems are mainly generated by orbital interactions, and their intensities are closely related to the distance between atoms, shape, and orientation of molecular orbitals. We constructed the computational model with S⋯O NoCL (Fig. 4a and S5) to investigate the functional relationship between $E^{(2)}$ and the length of NoCL $d$(S⋯O), the angle between S⋯O NoCL and S-C bond ($\alpha$), and the dihedral angle between two thiophene rings ($\theta$) which is closely related to molecular structural planarity and disorder[57]. It is found the $E^{(2)}$ as an exponential function of the distance, and the cosine function of the angle $\alpha$ or $\theta$. Based on this, we proposed a simple descriptor that contains several geometrical parameters (Fig. 4a),

$$S = (-\cos\alpha) \cdot \cos^2\theta \cdot (1 - e^{\Delta d})^2 \tag{1}$$

To verify the descriptor, we plot the $E^{(2)}$ versus $S$, which are calculated based on the optimized equilibrium geometries for 36 isolated systems in Fig. 4b and Table S2. Impressively, the linear fitting coefficient $R^2$ is as high as 0.927, which indicates a very strong positive correlation between $E^{(2)}$ and $S$. Moreover, the $S$ value varies in the order of S⋯O < Se⋯O < Te⋯O and S⋯F < Se⋯F < Te⋯F, which is in good agreement with the order of strength of NoCLs obtained above by $E^{(2)}$. This is mainly because from S to Te the atomic radius gradually increases while the distances of X⋯Y and other factors are almost unchanged, which results in the increase of $\Delta d$, corresponding to the strengthening of NoCLs in this order. In addition, as it is described above that the orbital interactions of hydrogen bond in **PhM-H**

compounds happen between s-orbital of H atom and p- or sp²-orbital of Y atoms, and it is almost independent of the orientation of C-H bond because s-orbital is spherical. And if removing cosα from the $S$ for **PhM-H** systems, the results would become better (Figure S6).

Considering the effect of the basis set and method on the results, the $E^{(2)}$ and $S$ were further calculated for 11 compounds of 36 with strong NoCLs at B3LYP(D3)/def2-TZVP, ωB97XD/def2-TZVP and CCSD(T)/def2-TZVP levels (Fig. 4c, d). The linear correlation between $E^{(2)}$ and $S$ becomes stronger at large basis set and higher theory levels. Based on the optimized geometries by B3LYP(D3)/def2-TZVP, the linear fitting coefficient is improved from $R^2 = 0.974$ at B3LYP(D3)/def2-TZVP level to $R^2 = 0.991$ at CCSD(T)/def2-TZVP level. Based on the optimized geometries by ωB97XD/def2-TZVP (Figure S7), the CCSD(T)/def2-TZVP gives the best linear correlation between $E^{(2)}$ and $S$ with $R^2 = 0.992$ (Fig. 4d). In addition, the linear correlation between $E^{(2)}$ and $S$ of 56 compounds is still strong with $R^2 = 0.907$ (Figure S8). Therefore, $S$ is an excellent descriptor to characterize the strength of NoCL, and it can be roughly judged that there is an effective NoCL when its value is larger than ca. 0.14 (corresponding to a significant $E^{(2)}$ less than ca. −1.00 kcal mol⁻¹).

As it is well known that reorganization energy or relaxation energy is an important physical parameter to character the optoelectronic properties. Reorganization energy is defined as the energy dissipated from the equilibrium geometry in the initial state to the relaxed equilibrium geometry in the final state for an isolated molecule (see Figs. 5a and 4c). For the photo-excited electron transition processes between the potential energy surfaces of the ground state (GS) and the excited state (ES) in Fig. 5a, when one molecule at a stable point (**a**)

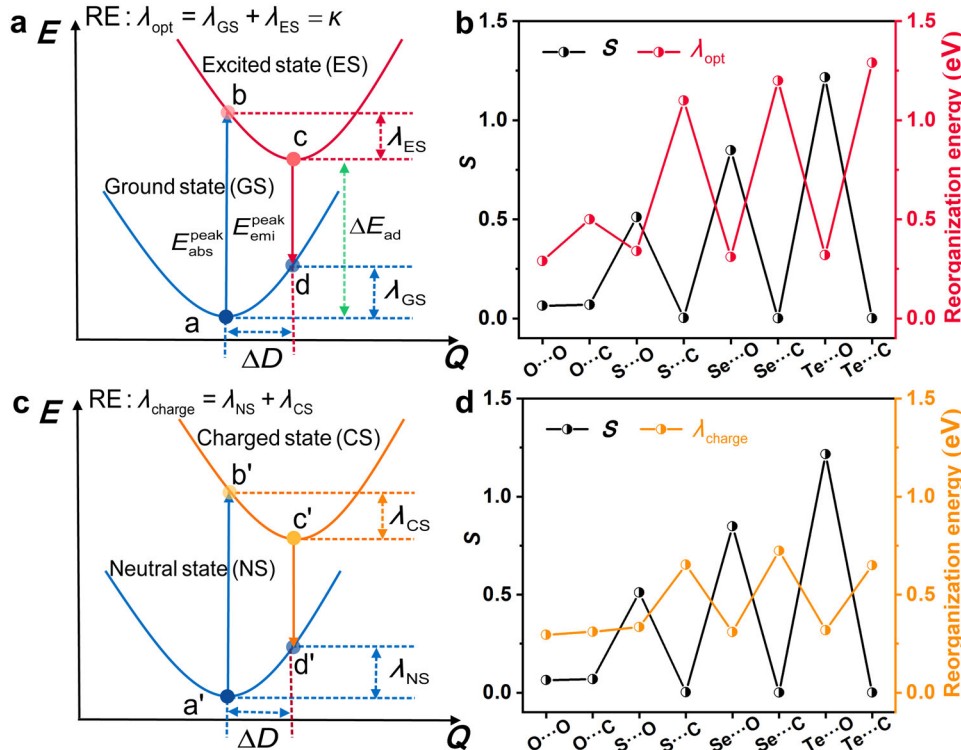

**Fig. 5 | The relationship between reorganization energy and descriptor. a** The photo-excited electron transition processes between the potential energy surfaces of the ground state (GS) and the excited state (ES) in organic molecules with displacement $\Delta D$, reorganization energy $\lambda_{ES}$ and $\lambda_{GS}$, and the Stokes shift $\kappa$. **b** The $S$ versus the optical reorganization energy of **PhM**(X···O) and **PhM**(X···C). **c** For electron-pumped charge transfer processes between the neutral (ground) state (NS) and the charged states (CS) in organic molecules with corresponding displacement $\Delta D$, reorganization energy $\lambda_{NS}$ and $\lambda_{CS}$. **d** The $S$ versus the charged reorganization energy of **PhM**(X···O) and **PhM**(X···C).

absorbs a phonon to form an excited state (**b**), the excited molecule firstly experiences a relaxation of geometrical structure to a new equilibrium point (**c**) by releasing reorganization energy ($\lambda_{ES}$) and then decays to the ground state (**d**) by radiative and nonradiative processes, finally relax to the initial ground-state equilibrium point (**a**) by giving out reorganization energy ($\lambda_{GS}$). The reorganization energy decides the shape and width of the optical spectra. Under the framework of the linear coupling model, the sum of the two reorganization energies can be crudely considered as the Stokes shift between the absorption and emission spectra, $\kappa = \lambda_{GS} + \lambda_{ES}$[58,59]. Moreover, there is a positive relationship between the nonradiative decay rate and the excited reorganization energy, in which the excitation energy is always much larger than the reorganization energy[60, 61]. Consequently, small reorganization energy can effectively block the nonradiative decay process. For electron-pumped charge transfer processes, the charge transfer rates between molecule dimers can be calculated by Marcus rate equation in the high temperature and short time limits, $k_{et} = A\exp\left[-\frac{(\Delta G + \lambda_{charge})^2}{4\lambda_{charge}k_B T}\right]$. For the charge transfer among the same molecules ($M + M^+ \rightarrow M^+ + M$), $\Delta G$ can be approximately zero, and there is an inverse relationship between the rate constant of charge transfer and the charge reorganization energy $\lambda_{charge}$. As a result, the molecules with small reorganization energy usually are designed to exhibit excellent charge transport property[62]. The $\lambda_{charge}$ can be calculated by the sum of $\lambda_{NS}$ and $\lambda_{CS}$ between the neutral (ground) state (NS) and the charged states (CS) in organic molecules, as shown in Fig. 5c.

The optical reorganization energy $\lambda_{opt}$ and charge reorganization energy $\lambda_{charge}$ of **PhM**(X···O) and **PhM**(X···C) with X = O, S, Se, and Te were calculated as shown in Fig. 5b, d, and Table S3. As shown in Fig. 5b, the $S$ behaves good negative relationship with $\lambda_{opt}$ (Stokes shift $\kappa$). The Stokes shifts of the systems with strong NoCLs are far smaller

than those of the corresponding systems without NoCLs. Moreover, the larger $S$ is, the larger the difference of Stokes shift is. For example, the Stokes shift from **PhM**(Se • ••C) ($S = 0.02$, $\kappa = 117$ nm) to **PhM**(Se • ••O) ($S = 0.85$, $\kappa = 42$ nm) is decreased by 75 nm, while that from **PhM**(Te • ••C) ($S = 0.01$, $\kappa = 164$ nm) to **PhM**(Te • ••O) ($S = 1.22$, $\kappa = 46$ nm) decreased by 118 nm. The linear correlation between $\Delta\lambda_{opt}$ and $S$ is very good with $R^2 = 0.801$ for 6 pairs of compounds of type II molecules with apparent noncovalent interactions (Figure S9). Thus far, the relationship among the molecular structure, the NoCL strength, and the optical properties was well established. Compared with the counterpart $\lambda_{opt}$, $\lambda_{charge}$ is relatively smaller, which suggests the molecules in charged states are more rigid than those in the excited states. Still, the $S$ behaves a good negative relationship with $\lambda_{charge}$, as shown in Fig. 5d, which indicates that $S$ is able to effectively tune the charge transport property of an organic semiconductor.

### Experimental validation of noncovalent interactions
To confirm the theoretical prediction, eight compounds **PhM** (X···Y) of X = O, S, Se, and Te, and Y = O and C were synthesized (Figure S11–S18), and crystals of six compounds were cultured for single crystal X-ray diffraction (Fig. 6). The conformations of all six compounds reproduced the theoretical predictions as shown in Fig. 3. Specifically, only one conformation was observed for **PhM-H**(O···O), **PhM**(S···O) **PhM**(Se···O), and **PhM**(Te···O). In these compounds, the $d$ values are 2.38(9) Å for H···O, 2.68(2) Å for S···O, 2.69(1) Å for Se···O, and 2.76(0) Å for Te···O, respectively, which are much shorter than the corresponding $dv$, confirming the existence of strong NoCLs. Differently, both **PhM/PhM-H** (S···C) and **PhM/PhM-H** (Se···C) possess two conformations, which fully prove the theoretical predictions above. Moreover, the torsion angles between chalcogenide and phenyl rings in **PhM**(S···O) (4.68°, 5.59°) and **PhM**(Se···O) (2.54°, 5.32°) are much

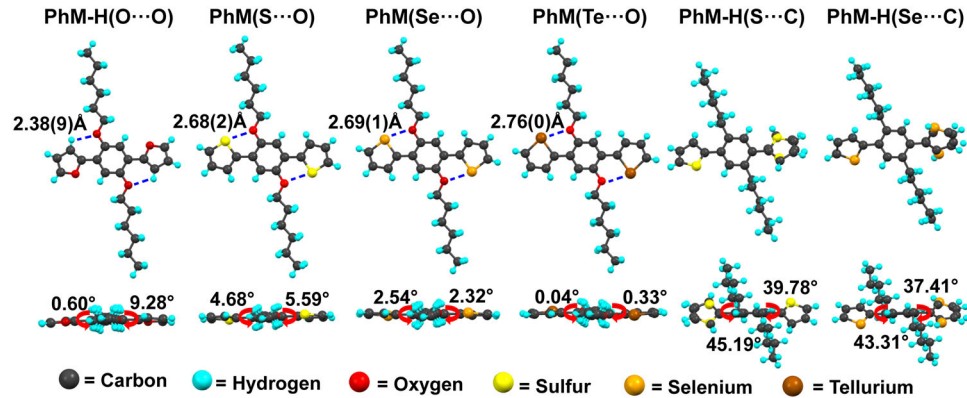

**Fig. 6 | The single crystal structures of six compounds.** Key distances between atoms and dihedral angles are marked with blue dot lines and red arrows respectively, in top view (up) and side view (down).

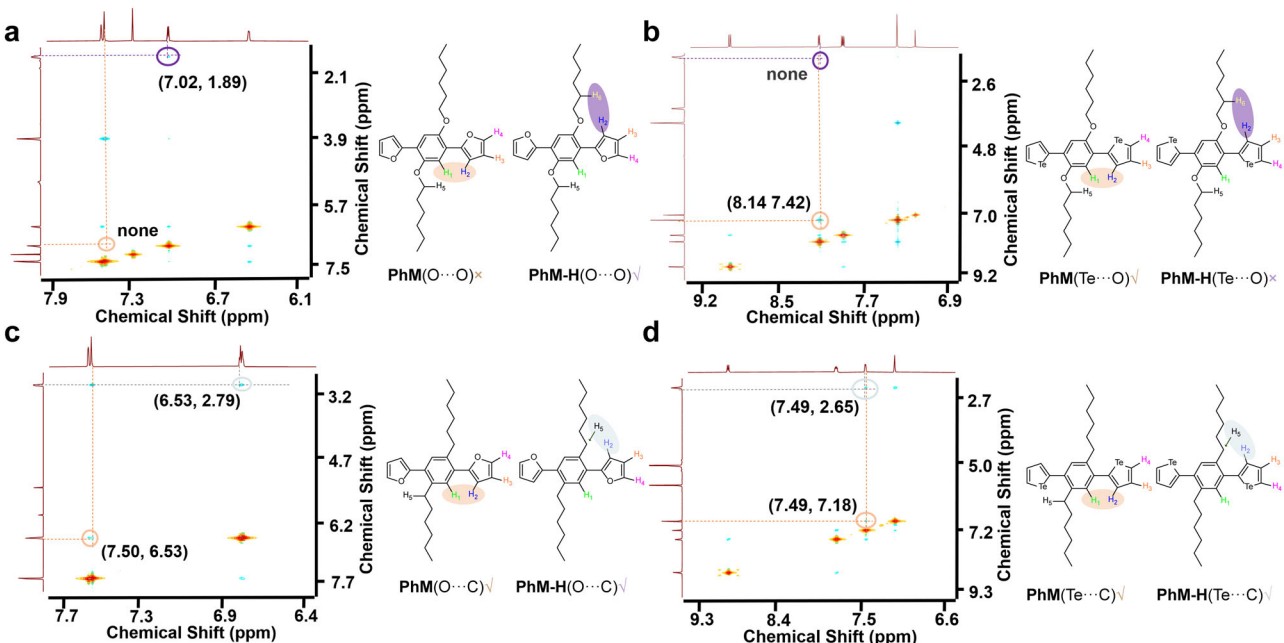

**Fig. 7 | NOESY ¹H-¹H NMR spectra of compounds in *d2*-tetrachloroethane. a PhM-H(O···O), b PhM(Te···O), c PhM-H(O···C), and d PhM(Te···C).** Relevant NOESY interaction signals are circled as brown between $H_2$ (marked as blue) and $H_1$ (marked as green) in **PhM**(X···Y) (X = O, Te; Y = O, C); purple between $H_2$ (marked as blue) and $H_6$ (marked as yellow) in **PhM-H**(O···O) and **PhM-H**(Te···O); and silver between $H_2$ (marked as blue) and $H_5$ (marked as black) in **PhM-H**(Te···C) and **PhM-H**(O···C). Coordinates of the signals are displayed in black brackets, and missing signals are shown as "none".

smaller than those in **PhM**(S···C) (45.19°, 39.78°) and **PhM**(Se···C) (43.31°, 37.41°), illustrating the "lock of planarity" function of NoCLs.

The ¹H-¹H NOESY NMR of eight compounds was measured in *d2*-tetrachloroethane solution, which is always used to certify conformational preferences through spatial interactions between different ¹H signals (Fig. 7a and Figure S19)[63]. As shown in Fig. 7a, the cross-peak is observed between $H_6$ and $H_2$, but misses between $H_1$ and $H_2$, which demonstrates that **PhM-H**(O···O) is more stable than **PhM**(O···O) in *d2*-tetrachloroethane and the O···H NoCL is stronger than O···O one. Similarly, the cross-peak between $H_1$ and $H_2$ in Fig. 7b suggests stable **PhM**(Te···O) conformation with strong Te···O NoCLs. Both the cross-peaks between $H_5$ and $H_2$, and between $H_1$ and $H_2$ are observed in Fig. 7c, d, which indicates two conformations for **PhM**(O···C) and **PhM-H**(O···C), and **PhM**(Te···C) and **PhM-H**(Te···C) coexist owing to freely rotatable aromatic rings without NoCLs, suggesting no effective NoCLs in these compounds. These NMR observations are consistent with the experimental results that no single conformation was observed for

**PhM-H**(S···C) and **PhM-H**(Se···C) in their respective crystals as shown in Figure S19. Nevertheless, **PhM**(S···O) seems to be inconsistent with its crystal structure since it exhibits no cross-peaks neither between $H_6$ and $H_2$, nor between $H_1$ and $H_2$ (Figure S19). According to the geometry structure of **PhM**(S···O) calculated in Table S2, it is found that the $\theta$ of isolated **PhM**(S···O) is calculated to be 17.29°, which is larger than that of 4.68°/5.59° in crystal, implying a more twisted conformation in solution. Since the strengths of NoCLs decrease sharply from $\theta = 0°$ to 20° (Fig. 2), the S···O NoCL of **PhM**(S···O) may become very weak in solution, causing the alienation of $H_1$ and $H_2$. Thus, these experimental observations reinforce the reliability of theoretical predictions. In addition, The ⁷⁷Se and ¹²⁵Te NMR of **PhM**(Se···O), **PhM**(Se···C), **PhM**(Te···O), and **PhM**(Te···C) are further measured and the results were shown in Figure S20. The ⁷⁷Se NMR of **PhM**(Se···O) displays one single peak of symmetric Se nucleus located at 645.70 ppm, with a downfield shift of 9.15 ppm compared to the signal of **PhM**(Se···C) at 636.55 ppm. Since the inductive effect of the O atom generally decays

**Table 1 | The calculated photophysical properties of the new synthesized eight systems, as well as experimental values in parentheses**

| Systems | $\lambda_{abs}$ (nm) | $\lambda_{em}$ (nm) | $\kappa$ (nm) | $\Delta\kappa$ (nm) |
|---|---|---|---|---|
| **PhM-H**(O⋯O) | 359 (353[a], 380[b]) | 392 (399[a], 425[b]) | 33 (46[a], 45[b]) | 17 (24[a], 40[b]) |
| **PhM-H**(O⋯C) | 328 (305[a], 310[b]) | 378 (375[a], 395[b]) | 50 (70[a], 85[b]) | |
| **PhM**(S⋯O) | 376 (362[a], 403[b]) | 420 (399[a], 444[b]) | 44 (37[a], 41[b]) | 61 (63[a], 10[b]) |
| **PhM**(S⋯C) | 297 (285[a], 324[b]) | 402 (385[a], 375[b]) | 105 (100[a], 51[b]) | |
| **PhM**(Se⋯O) | 391 (376[a], 409[b]) | 433 (410[a], 439[b]) | 42 (34[a], 31[b]) | 75 (66[a], 26[b]) |
| **PhM**(Se⋯C) | 294 (289[a], 329[b]) | 411 (399[a], 385[b]) | 117 (100[a], 56[b]) | |
| **PhM**(Te⋯O) | 401 (390[a], 428[b]) | 447 (425[a], 460[b]) | 46 (35[a], 32[b]) | 118 (91[a], 73[b]) |
| **PhM**(Te⋯C) | 324 (290[a], 295[b]) | 488 (416[a], 400[b]) | 164 (126[a], 105[b]) | |

[a], Solution; [b], Film.

along chemical bonds and disappears after three δ bonds, this phenomenon can only be triggered by Se⋯O interaction. Note that this trend is in accordance with that in literature reported by Tomita[64]. Similarly, the downfield shift phenomenon is observed in **PhM**(Te⋯O) (851.62 ppm) in comparison to that of **PhM**(Te⋯C) (841.78 ppm) in [125]Te NMR, which reveals the existence of Te⋯O interactions. Overall, the reflected results by [1]H-[1]H NOESY NMR tell us that the Te⋯O, O⋯H, and S⋯O are significant NoCLs while the O⋯C, Te⋯C, and H⋯C are not effective NoCLs in these compounds, which are consistent with the theoretically predicted results.

In order to reveal the nature of NoCLs, the effect of solvents on NoCLs was investigated. Taking **PhM**(Te⋯O) with strong NoCLs as an example, the [1]H-[1]H NOESY NMR is further measured in d2-dichloromethane (d2-DCM) and d6-dimethylsulfoxide (d6-DMSO) shown in Figure S21. From Figure S21 and Fig. 7b, it can be seen that the obvious cross-peak between H₁ and H₂ appears in the three solvents. The position of the cross-peak hardly shifts with the increase of the polarity of the solvent, which implies these NoCLs are not controlled by the electrostatic interactions[47].

To prove the prediction of the relationship between the descriptor and the Stokes shift, the absorption and emission spectra of the eight compounds were performed by theoretical calculations and experimental measurements, and the results are summarized in Table 1 and Figure S22–S23. In Table 1, it was observed that the calculated results of isolated molecules agreed well with the experimental results in solution, regardless of the changing trend or numerical values, including the absorption peak, emission peak, and Stokes shift. Compared with **PhM**(X⋯C) (X = S, Se, Te), there display immense redshifts of the absorption peaks in **PhM**(X⋯O) (X = S, Se, Te), which accounts for the introduction of NoCLs for higher molecular planarity[65]. The resultant Stokes shift sharply decreased from **PhM**(X⋯O) to **PhM**(X⋯C) (X = S, Se, Te), which depicts the enhancement of rigidity brought by NoCLs. Specific $\Delta\kappa$ values are rendered to estimate the relative ability of NoCLs in altering rigidity with 61 (63) nm for S⋯O interaction, 75 (66) nm for Se⋯O interaction, and 118 (91) nm for Te⋯O interaction. This implies the stronger the NoCLs, the more its ability in enhancing rigidity. Moreover, the intensity order of S⋯O < Se⋯O < Te⋯O of the NoCLs is perfectly reflected by the $\Delta\kappa$ values, which are in good agreement with the theoretically calculated results above. The same experiments are carried out in tolune and THF solvents with different polarity, and the Stokes shifts and their changes exhibit extremely small changes (Table S6). These indicates that the NoCLs of these systems are not controlled by electrostatic interactions. The same change trend and feature of the spectral properties are found in aggregates for the systems. Therefore, it is safe to say that there is an inverse correlation between the descriptor and Strokes shift, and the luminescent properties may be regulated through changing the S of NoCL in organic molecules.

To extend the application of this descriptor, twelve compounds were selected to be investigated (Fig. 8a). First, we calculated the

theoretical S ($S_{Theo}$) based on the theoretical equilibrium structures in the gas (solid) phase and the experimental S ($S_{Exp}$) based on the experimental crystal structures for the six compounds (Fig. 8b and Table S4). Excitingly, the $R^2$ of the solid phase is higher with a value of 0.995 than that of the gas phase (0.982), which demonstrates that the $S_{Exp}$ based on the crystal structure is also very reliable. To confirm the assumption, we arbitrarily select a series of compounds from the CCDC crystal database to calculate $E^{(2)}$ and the $S_{Exp}$ based on experimental crystal structures and plot them in Fig. 8c and Table S5. Significantly, the resultant linear fitting coefficient $R^2$ is very high with a value of 0.972, which reveals that the S can be applied widely. Thus far, it is safe to conclude that S is a suitable descriptor for evaluating the strength of NoCLs, which would be used for further machine-learning-based molecular screening studies.

Herein, 56 organic semiconductors with and without noncovalent interactions (X⋯Y, X = O, S, Se, Te; Y = C, F, O, S, Cl) were investigated to show that the interactions are mainly derived from orbital interactions based on the theoretical calculations and experimental results. The strength of interactions followed the order of S⋯F < Se⋯F < Te⋯F, S⋯O < Se⋯O < Te⋯O, and X⋯F < X⋯O (X = S, Se and Te). In contrast, O⋯Y (Y = F, Cl, O, S), X⋯C (X = S, Se, Te), and Se⋯Y (Y = S, Cl) cannot form effective noncovalent interactions in the systems. Furthermore, it is disclosed that the strength of noncovalent interactions is closely related to the structural parameters, including the molecular planarity, the angle between two relevant orbitals, and the distance between the two related atoms. Significantly, a descriptor $S = (-\cos\alpha) \cdot \cos^2\theta \cdot (1 - e^{\Delta d})^2$ was built based on a few geometric parameters, which can be easily obtained without performing any complex quantum chemical calculations, and exhibited a strikingly positive correlation with the strength of noncovalent interaction. Moreover, it exhibits a good inverse correlation with the optical reorganization energy and charge reorganization energy, revealing the ability of noncovalent interaction to regulate the luminescent and charge transport properties of organic semiconductors. Importantly, the single crystal structures, [1]H, [77]Se, and [125]Te NMR, and UV-vis spectra of the eight new synthesized systems fully confirmed the theoretical predictions. This work provides an in-depth understanding and simple description of noncovalent interaction, which is important for the rational molecule design of high-performance organic/polymeric semiconductors.

## Methods
### Computational details
All the geometrical and electronic structures of the investigated system in the ground state were calculated at B3LYP(D3) or ωB97XD level[66] of theory using the basis set 6-31 + G(d) (C, H, O, F, Cl) and LANL2DZ for Se and Te or def2-TZVP in Guassian16 Program[67], including geometrical parameters, energy, atomic dipole moment-corrected Hirshfeld population (ADCH) charge analyses[66,68], natural bond orbital (NBO) analysis[54]. NBO analysis have been carried out using Multiwfn software[69]. The CCSD/def2-TZVP method was used to

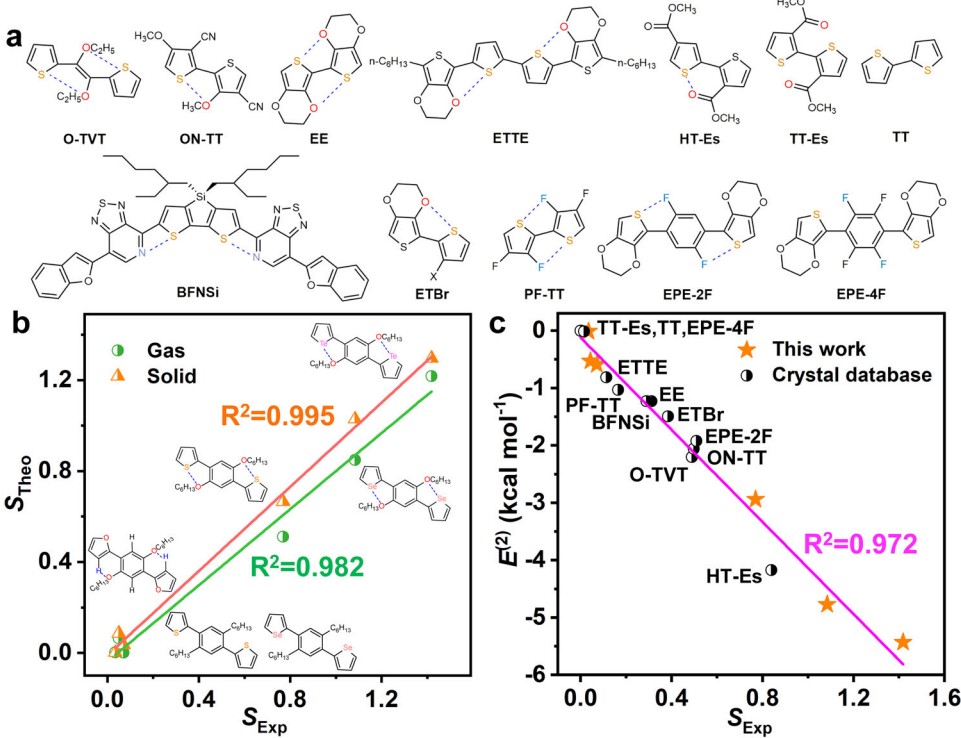

**Fig. 8 | The application of the descriptor in reported compounds. a** Molecular structures of the compounds from experimental crystal database. **b** The comparison between $S_{Theo}$ and $S_{Exp}$ in the gas phase and solid phase with fitted lines for newly synthesized six systems. **c** The positive correlation between $E^{(2)}$ and $S$ with fitted line from experimental crystal structures.

calculate the orbital interaction. As for the solid phase, the initial structure is obtained from the X-ray crystal structure, and geometrical structures in the ground state are calculated based on QM/MM model through Gaussian16 Program. The centered molecule is treated as a high layer and calculated at B3LYP(D3)/def2-TZVP level for QM, and the surrounding molecules are treated as a low layer and computed by MM with UFF forces field. The reorganization energy is calculated using the adiabatic potential energy surface method in MOMAP[70–72].

### General synthesis of PhM(X⋯O)/PhM(X⋯C)

Preparations of 1,4-dihexyl-2,5-diiodobenzene, 1,4-dihexyloxy-2,5-diiodobenzene, 2-tributylstannyl-derivatives are illustrated thoroughly in Supplementary Information. To a round-bottom flask (100 mL), 1,4-dihexyl-2,5-diiodobenzene (1,4-dihexyloxy-2,5-diiodobenzene) (1.0 equiv), 2-tributylstannyl-derivatives (2.5 equiv) and Pd(PPh₃)₄ (5% equiv) were added under nitrogen in anhydrous toluene (20.0 mL). The mixture was stirred and refluxed at 120 °C for 12 h. Afterwards, an aqueous potassium fluoride solution (3.0 M, 40.0 mL) was added to the mixture. After quenching the reaction, the organic phase was extracted with dichloromethane (3 × 100 mL). The combined organic portion was collected, washed with water and brine, and dried over MgSO₄. After filtration, the solution was filtered through a short silica gel column (petroleum ether), concentrated to afford the crude product, which is subjected to the recrystallization in hexane. Finally, the crystals were dried in vacuo to afford **PhM**(X⋯O)/**PhM**(X⋯C). The detailed synthesis is described in Supplementary Information, and the final products were characterized by ¹H-NMR, ¹³C-NMR, ⁷⁷Se-NMR, and ¹²⁵Te-NMR. (Supplementary Section 13).

### UV-vis absorption and photoluminescence spectra

UV-Vis absorption spectra were measured on a Gary 60 UV-Vis Spectrophotometer. Photoluminescence spectra were measured on a FLS 1000 (EDINBURGH INSTRUMENTS) with an Xenon Lamp. All liquid samples were well dissolved in chloroform. All film samples were spin-coated on glass substrates.

### General crystallization of single-crystalline PhM(X⋯O)/PhM(X⋯C)

A glass vial (5 mL) containing a chloroform or toluene solution of the compounds (2 mg) was placed inside a vial (20 mL) containing methanol. After 2–7 days, white (yellow) solid single crystals were collected.

## Data availability

The single-crystal structure of **PhM**(X⋯O)/**PhM**(X⋯C) is archived at the Cambridge Crystallographic Data Centre under the reference number CCDC-2152134, 2152135, 2152136, 2152139, 2152140, and 2152141. These data can be obtained free of charge from The Cambridge Crystallographic Data Centre via www.ccdc.cam.ac.uk/data_request/cif. Crystal data are available in Supplementary Data 1–6. Atomic coordinates for the optimized geometries of the studied systems are available in Supplementary Data 7. The authors declare that all the data supporting the finding of this study are available within this article and its Supplementary Information files and are available from the corresponding author on request.

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

## Acknowledgements

This work is supported by the National Natural Science Foundation of China (51925306, 22273105, 21973099, and 52120105006), National Key R&D Program of China (2018FYA 0305800), and the Fundamental Research Funds for the Central Universities. We gratefully acknowledge HZWTECH for providing computation facilities.

## Author contributions

H.H. and Q.P. conceived and supervised the study. M.L. conducted the theoretical calculations, analyzed data, and prepared the manuscript. X.H. conducted the experimental part and analyzed experimental data. H.C. synthesized (PhM (Te···O)) compound. M.L. and X.H. made equal contributions to this study.

## Competing interests

The authors declare no competing interests.
