## [Peer Review File · Nature Communications]

A molecular descriptor of intramolecular noncovalent interaction for regulating optoelectronic properties of organic semiconductorsReviewers' Comments:

Reviewer #1:

Remarks to the Author:

Intramolecular noncovalent interaction has become an efficient method to tune the physicochemical properties of organic/polymeric semiconductors in experiments. However, there is a general lack of direct quantitative relationship between molecular geometry, strength of noncovalent interaction and optoelectronic property. In this contribution, Huang and coworkers have performed theoretical calculations on 56 molecules with and without noncovalent interaction, and proposed a descriptor S based on simple structural parameters which demonstrates excellent positive correlation with noncovalent interaction energy. Moreover, the descriptor exhibits a good inverse correlation with the reorganization energies of the photo-excited states or electron-pumped charged states in organic/polymeric semiconductors. These properties of the descriptor are strongly supported by experimental crystal structures, NMR, and optical spectra. The structure-property relationships determined in this work are very helpful for rational molecule-design and screening of high-performance organic/polymeric semiconducting materials. I believe this contribution is important for both the chemistry and materials community, thus I strongly recommend it for publication after addressing the following points.

1. Does the NoCLs include hydrogen bonds? It is well known that hydrogen can exhibit strong electrostatic interaction, but orbital interactions dominate in your system. Please give the detailed comparison and discussion.
2. The effect of solvents on noncovalent interactions should be described more, for example, what about the $\Delta\kappa$ in different solvents.
3. In line 277, "van der Waals radium" is a typo, and here needs to use abbreviation "dv" as defined above.
4. In line 321, the order of reference number and full stop is different from others.
5. In Figure 6c, the unit "kcal/mol" lacks brackets.
6. In Figure 1a and 1d, sans serif "Å" should be used.

Reviewer #2:

Remarks to the Author:

This manuscript presents a study of intramolecular noncovalent interactions and their implications in the stability and optoelectronic properties of organic semiconductors. A set of 56 molecules involving different interactions is studied using ab initio methods such as density functional theory and coupled cluster theory (the chemically distinct molecules are actually 36 but 20 of them are considered in two different stable structures). Out of the most stable structures, 15 are predicted to have a significant noncovalent conformational lock (NoCL) character. A descriptor based exclusively on geometric parameters is then introduced to predict and quantify the NoCL character. The correlation of this descriptor with the optical properties of the polymers is also demonstrated. Finally, the theoretical predictions and the quality of the descriptor are validated by performing experiments on a subset of the molecules.

This is a beautiful study that mixes computational DFT characterization, determination of a simple and efficient geometric descriptor, and experimental validation. The coupling of the descriptor with machine learning models could be useful in the future to systematically screen and design new polymers. Because of its potential impact I endorse its publication in Nature Communications.

Before publications there are two main issues that the authors should address:

-The presentation of the paper is certainly rigorous but rather involved. For example, Sec. 2.1 is very detailed and tedious to read; this is suitable for a more technical journal but for publication in Nat. Comm. the main messages should come across in a more straightforward way, leaving additional details for supplementary information. While the paper is high quality from the scientific point of view,

I think that the presentation is cumbersome in parts of it.

-On the more technical side I have the following concerns: The descriptor S is judged to be "an excellent descriptor to characterize the strength of NoCL". However, S is not capable to characterize the PhM-H conformations that in figure 3(b) correspond all to a similar small value of this descriptor. However, at least two of these molecules are considered to have significant NoCL character. This point should be clarified.

Reviewer #3:

Remarks to the Author:

Liu and co-workers present a joint computational and experimental work with the aim of developing a descriptor to determine the structure of pi-conjugated molecules that are of interest for organic semiconductors. Notably, the descriptor, S , is very similar to a descriptor already proposed by Che and Perepichka (Angewandte Chemie, 2020, DOI: 10.1002/anie.202011521), though no citation for the work is given. Further, the authors appear to conflate intramolecular covalent (i.e., orbital overlap) and non-covalent (i.e., electrostatics, dispersion, etc.) interactions in their descriptor, though the title and basis of the work is focused on "intramolecular noncovalent interactions"; if the authors do not want to conflate these items, a definition of when a bond is formed or not should be provided, especially since the orbital term seems to play a significant role. There are also several concerns regarding the physical understanding provided, given many works in this field that provide clear delineations of these effects.

Comments for author consideration:

Recent work from Karunasena and co-workers (Chem. Mater., 2021, DOI: 10.1021/acs.chemmater.1c02335) suggests rather that pi delocalization across the carbon backbone is more important than intramolecular non-covalent interactions in determining conformations. It would be interesting for the authors to evaluate this aspect. In addition to this work, Kharandiuk and co-workers (Chem. Mater., 2019, DOI: 10.1021/acs.chemmater.9b01886) also use atoms-in-molecules (AIM) theory to examine similar effects discussed in this paper.

From a computational perspective, sufficient information is provided to generally repeat the calculations. However, there are several questions pertaining to the methods deployed. First, B3LYP is well established to over-delocalize wavefunctions, which will impact the orbital analyses carried out (especially when the identifier is about 0.1 Ang). Further, B3LYP tends to favor CT-like interactions, which may also influence the descriptions. Hence, it is not clear how dependent the analyses are on a functional that suffers greatly from multi-electron self-interaction errors (MESIE). It is suggested that the authors make use of range-separated hybrid functionals to correct for MESIE. (Note that the CCSD(T) results are probably also influenced by this error given that the geometry used comes from the DFT calculations.)

All dihedral scans should be carried out from 0 to 180 degrees. For the systems that only went to 90 degrees, a secondary minimum should also be present.

kT at room temperature is 0.6 kcal/mol. It is difficult to say that one well is much more stable than the other when comparing the minima. Rather, what is important is the barrier to rotation, as the intramolecular interactions tend to be very weak.

The authors' discussion on S—O interactions in PhM and PhM-H appears to be number-crunching. The authors should provide a rationale for why orbital interactions are strong.

"Strength of NoCL exhibits the order of $S \cdots F < Se \cdots F < Te \cdots F$, $S \cdots O < Se \cdots O < Te \cdots O$, and $X \cdots F < X \cdots O$ ($X=S$,

Se and Te)". An explanation for the observed trend would be beneficial.

Page 9, line 177. The molecules in this investigation should have sp^3 and sp^2 hybrid orbitals for an organic chemist. The authors should clarify the mention of 'hybrid sp-orbital'.

While the experiments confirm theoretical findings, the dataset is small to generalize the findings. The authors should expand the dataset by generating the S values and E_2 for all structures used in estimating the PES. This should be more conclusive that the S descriptor performs well for all geometries, not just equilibrium geometry

The discussion of the reorganization energy is confusing. The molecules are not expected to undergo significant changes in twisting as they are already close to planar. Analyses of the vibrational modes (Huang-Rhys factors, etc) will show that most of the reorganization is due to the changes in carbon-carbon bond lengths, as is well established for pi-conjugated systems. So, it is not clear that that the correlation says anything about the systems under investigation.

The comparison of the TDDFT results with the experimental UV/vis shows that the optimized geometries are good estimates of conformations available in solution. There really is no comparison to be made to the proposed descriptor.

Why is the analysis of descriptor S for optical reorganization energy restricted to only PhM systems and PhM(O--O)? The authors should perform the analysis for all 56 molecules for completeness.

The S descriptor seems reasonable for E_2 value estimation. However, the descriptor does not predict the trend in the values for estimating the reorganization energy. For instance, in Figure 4b, the S values increase from 0.5 to 1.5, but the reorganization energy is still close to 0.3 eV. Authors should provide more evidence/data to show the actual applicability of the descriptor, which now appears bleak.

Additional minor comments:

In figure 1f, what do the two green traces represent? There is only one Elec legend.

Figure 5 atom color coding for 3D molecular representation should be included in the figure caption to improve readability.

Figure 5: The authors do not show the crystal structure. Should rephrase it along the lines – of "molecular geometry from the single crystal structure."

F in equation 1 of Chart S1 in SI is not defined

Abstract line 43: typo 'predication'

Typo' van der Waals radius' throughout the manuscript and SI.

Page 12, line 225 'As it well known' something is missing in the phrase.

Page 13, line 241: Typo "nonradaitive"

Reviewer #4:

Remarks to the Author:

Liu and co-workers present a joint computational and experimental work with the aim of developing a descriptor to determine the structure of pi-conjugated molecules that are of interest for organic

semiconductors. Notably, the descriptor, S , is very similar to a descriptor already proposed by Che and Perepichka (Angewandte Chemie, 2020, DOI: 10.1002/anie.202011521), though no citation for the work is given. Further, the authors appear to conflate intramolecular covalent (i.e., orbital overlap) and non-covalent (i.e., electrostatics, dispersion, etc.) interactions in their descriptor, though the title and basis of the work is focused on "intramolecular noncovalent interactions"; if the authors do not want to conflate these items, a definition of when a bond is formed or not should be provided, especially since the orbital term seems to play a significant role. There are also several concerns regarding the physical understanding provided, given many works in this field that provide clear delineations of these effects.

Comments for author consideration:

Recent work from Karunasena and co-workers (Chem. Mater., 2021, DOI: 10.1021/acs.chemmater.1c02335) suggests rather that pi delocalization across the carbon backbone is more important than intramolecular non-covalent interactions in determining conformations. It would be interesting for the authors to evaluate this aspect. In addition to this work, Kharandiuk and co-workers (Chem. Mater., 2019, DOI: 10.1021/acs.chemmater.9b01886) also use atoms-in-molecules (AIM) theory to examine similar effects discussed in this paper.

From a computational perspective, sufficient information is provided to generally repeat the calculations. However, there are several questions pertaining to the methods deployed. First, B3LYP is well established to over-delocalize wavefunctions, which will impact the orbital analyses carried out (especially when the identifier is about 0.1 Ang). Further, B3LYP tends to favor CT-like interactions, which may also influence the descriptions. Hence, it is not clear how dependent the analyses are on a functional that suffers greatly from multi-electron self-interaction errors (MESIE). It is suggested that the authors make use of range-separated hybrid functionals to correct for MESIE. (Note that the CCSD(T) results are probably also influenced by this error given that the geometry used comes from the DFT calculations.)

All dihedral scans should be carried out from 0 to 180 degrees. For the systems that only went to 90 degrees, a secondary minimum should also be present.

kT at room temperature is 0.6 kcal/mol. It is difficult to say that one well is much more stable than the other when comparing the minima. Rather, what is important is the barrier to rotation, as the intramolecular interactions tend to be very weak.

The authors' discussion on S—O interactions in PhM and PhM-H appears to be number-crunching. The authors should provide a rationale for why orbital interactions are strong.

"Strength of NoCL exhibits the order of $S \cdots F < Se \cdots F < Te \cdots F$, $S \cdots O < Se \cdots O < Te \cdots O$, and $X \cdots F < X \cdots O$ ($X=S, Se$ and Te)". An explanation for the observed trend would be beneficial.

Page 9, line 177. The molecules in this investigation should have sp^3 and sp^2 hybrid orbitals for an organic chemist. The authors should clarify the mention of 'hybrid sp-orbital'.

While the experiments confirm theoretical findings, the dataset is small to generalize the findings. The authors should expand the dataset by generating the S values and E_2 for all structures used in estimating the PES. This should be more conclusive that the S descriptor performs well for all geometries, not just equilibrium geometry

The discussion of the reorganization energy is confusing. The molecules are not expected to undergo significant changes in twisting as they are already close to planar. Analyses of the vibrational modes (Huang-Rhys factors, etc) will show that most of the reorganization is due to the changes in carbon-

carbon bond lengths, as is well established for pi-conjugated systems. So, it is not clear that that the correlation says anything about the systems under investigation.

The comparison of the TDDFT results with the experimental UV/vis shows that the optimized geometries are good estimates of conformations available in solution. There really is no comparison to be made to the proposed descriptor.

Why is the analysis of descriptor S for optical reorganization energy restricted to only PhM systems and PhM(O--O)? The authors should perform the analysis for all 56 molecules for completeness.

The S descriptor seems reasonable for E2 value estimation. However, the descriptor does not predict the trend in the values for estimating the reorganization energy. For instance, in Figure 4b, the S values increase from 0.5 to 1.5, but the reorganization energy is still close to 0.3 eV. Authors should provide more evidence/data to show the actual applicability of the descriptor, which now appears bleak.

Additional minor comments:

In figure 1f, what do the two green traces represent? There is only one Eelec legend.

Figure 5 atom color coding for 3D molecular representation should be included in the figure caption to improve readability.

Figure 5: The authors do not show the crystal structure. Should rephrase it along the lines – of "molecular geometry from the single crystal structure."

F in equation 1 of Chart S1 in SI is not defined

Abstract line 43: typo 'predication'

Typo ' van der Waals radium' throughout the manuscript and SI.

Page 12, line 225 'As it well known' something is missing in the phrase.

Page 13, line 241: Typo "nonradaitive"

Point-by-Point Response to Referees

Reviewer: 1

Recommendation: I strongly recommend it for publication after addressing the following points.

Comments:

Intramolecular noncovalent interaction has become an efficient method to tune the physicochemical properties of organic/polymeric semiconductors in experiments. However, there is a general lack of direct quantitative relationship between molecular geometry, strength of noncovalent interaction and optoelectronic property. In this contribution, Huang and coworkers have performed theoretical calculations on 56 molecules with and without noncovalent interaction, and proposed a descriptor S based on simple structural parameters which demonstrates excellent positive correlation with noncovalent interaction energy. Moreover, the descriptor exhibits a good inverse correlation with the reorganization energies of the photo-excited states or electron-pumped charged states in organic/polymeric semiconductors. These properties of the descriptor are strongly supported by experimental crystal structures, NMR, and optical spectra. The structure-property relationships determined in this work are very helpful for rational molecule-design and screening of high-performance organic/polymeric semiconducting materials. I believe this contribution is important for both the chemistry and materials community, thus I strongly recommend it for publication after addressing the following points.

Our reply: Thanks for the reviewer's positive comments. We have revised carefully according to your opinions in the revised version.

RE: *Does the NoCLs include hydrogen bonds? It is well known that hydrogen can exhibit strong electrostatic interaction, but orbital interactions dominate in your system. Please give the detailed comparison and discussion.*

Our reply: Thanks for the reviewer's comments. Hydrogen bonds are one type of NoCLs,

which have important effects on the optoelectronic property in conjugated molecules [Chem. Rev. **117**, 10291-10318 (2017)]. In fact, the hydrogen bonds consist of electrostatic interactions and orbital interactions at the same time. The dominant one depends on the molecular structure of the studied systems [Monatshefte für Chemie - Chemical Monthly. **150**, 1267-1274 (2019)]. The orbital interactions through NBO analysis were computed to characterize the strength of hydrogen bonds [Molecules. **26**, 7622 (2021)], and strong orbital interactions between n(O) and the $\sigma^*(X-H)$ were reported in references [Sci. Rep. **6**, 34647 (2016)]. As shown in **Figure 1** in the main text, in **PhM-H(S \cdots O)**, the electrostatic interaction in hydrogen bond O \cdots H is extremely weak at $\theta=180^\circ$, and strengthened a little with the decrease of angle from 180° to 100° ; and the orbital interaction (n(O) $\rightarrow\sigma^*(H-C)$) sharply decreases from 180° to 140° and tends to zero at $\theta\in[180^\circ, 90^\circ]$. Thus, the orbital interaction is stronger than electrostatic interaction at $\theta\in[180^\circ, 160^\circ]$ and weaker than electrostatic interaction at $\theta\in[160^\circ, 90^\circ]$. In our systems, both the electrostatic interaction and orbital interaction of the hydrogen bonds are very weak.

RE: *The effect of solvents on noncovalent interactions should be described more, for example, what about the $\Delta\kappa$ in different solvents.*

Our reply: We really appreciate the reviewer's suggestion. Taking the suggestion, we have measured the $\Delta\kappa$ of the typical four **PhM** compounds in different solvents, and the results have been added in **Table S6** in supporting information. It is seen that the $\Delta\kappa$ exhibits very small change in different solvents, which is independent on solvent polarity. This excludes the electrostatic interactions to control the NoCLs. The related discussions have been added in the revised manuscript.

Table S6. κ and $\Delta\kappa$ in different solvents of chloroform, THF and toluene.

Systems	Chloroform		THF		Toluene	
	$\kappa(\text{nm})$	$\Delta\kappa(\text{nm})$	$\kappa(\text{nm})$	$\Delta\kappa(\text{nm})$	$\kappa(\text{nm})$	$\Delta\kappa(\text{nm})$
PhM(S \cdots O)	37	63	33	66	35	56

PhM(S...C)	100		99		91	
PhM(Se...O)	34		32		36	
		66		76		75
PhM(Se...C)	100		108		111	

RE: In line 277, “van der Waals radium” is a typo, and here needs to use abbreviation “*dv*” as defined above.

Our reply: Thanks for the reviewer’s comments. We have corrected the typo as “van der Waals radii” in the revised manuscript, which is highlighted with yellow color.

RE: In line 321, the order of reference number and full stop is different from others.

Our reply: Thanks so much for your instruction. We have corrected the typo in the revised manuscript, which now is highlighted with yellow color.

RE: In Figure 6c, the unit “kcal/mol” lacks brackets.

Our reply: We have added brackets in the revised version.

RE: In Figure 1a and 1d, sans serif “Å” should be used.

Our reply: Thanks so much for the reviewer’s advice. We have corrected the typeface in Figure 1a and 1d in the revised manuscript.

Reviewer: 2

Recommendation: Because of its potential impact I endorse its publication in Nature Communications.

Comments:

This manuscript presents a study of intramolecular noncovalent interactions and their implications in the stability and optoelectronic properties of organic semiconductors. A set of 56 molecules involving different interactions is studied using ab initio methods such as density functional theory and coupled cluster theory (the chemically distinct molecules are actually 36 but 20 of them are considered in two different stable structures). Out of

the most stable structures, 15 are predicted to have a significant noncovalent conformational lock (NoCL) character. A descriptor based exclusively on geometric parameters is then introduced to predict and quantify the NoCL character. The correlation of this descriptor with the optical properties of the polymers is also demonstrated. Finally, the theoretical predictions and the quality of the descriptor are validated by performing experiments on a subset of the molecules.

This is a beautiful study that mixes computational DFT characterization, determination of a simple and efficient geometric descriptor, and experimental validation. The coupling of the descriptor with machine learning models could be useful in the future to systematically screen and design new polymers. Because of its potential impact I endorse its publication in Nature Communications.

Our reply: We really appreciated the reviewer's strongly positive comments.

RE: *The presentation of the paper is certainly rigorous but rather involved. For example, Sec. 2.1 is very detailed and tedious to read; this is suitable for a more technical journal but for publication in Nat. Comm. the main messages should come across in a more straightforward way, leaving additional details for supplementary information. While the paper is high quality from the scientific point of view, I think that the presentation is cumbersome in parts of it.*

Our reply: Thanks so much for the reviewer's comments. Taking the reviewer's advice, the main messages have come across in a more straightforward way in the revised main text, and the additional details have been moved to supplementary information. For instance, the atomic dipole moment-corrected Hirshfeld (ADCH) charges have been moved from **Figure 1b** in main text to **Figure S1** in supplementary information, and the related discussions have been moved from main text to the supplementary information. Also, the detailed discussions of the evolution of PES are moved to the supplementary information.

Figure S1. The evolution of atomic dipole moment-corrected Hirshfeld (ADCH) charge as a function of the dihedral angle θ for (a) $M(S\cdots O)$ and (b) $PhM(S\cdots O)$.

RE: On the more technical side I have the following concerns: The descriptor S is judged to be “an excellent descriptor to characterize the strength of NoCL”. However, S is not capable to characterize the $PhM-H$ conformations that in figure 3(b) correspond all to a similar small value of this descriptor. However, at least two of these molecules are considered to have significant NoCL character. This point should be clarified.

Our reply: Thanks for the reviewer’s wonderful comments. We reanalyzed the feature of orbital interactions and the components of the descriptor. We found the orbital interactions of hydrogen bond in $PhM-H$ compounds happen between p- or sp^2 -orbital of heteroatoms and s orbital of hydrogen atom. Because the s orbital is almost spherical, the orbital interaction is independent on the bond angle (α) in computational model in **Figure 3a**. Removing $\cos\alpha$ from the descriptor and using $S = \cos^2 \theta \cdot (1 - e^{\Delta d})^2$ for $PhM-H$ systems, we obtained much better results, as shown in **Figure S5** in SI.

Figure S5. The correlation between $E^{(2)}$ and S for 36 compounds at B3LYP(D3)/6-31G+(d) level, which is removing $\cos\alpha$ from the S for $PhM-H$ systems.

Reviewer: 3 and 4

Comments:

Liu and co-workers present a joint computational and experimental work with the aim of developing a descriptor to determine the structure of pi-conjugated molecules that are of interest for organic semiconductors. Notably, the descriptor, S , is very similar to a descriptor already proposed by Che and Perepichka (Angewandte Chemie, 2020, DOI: 10.1002/anie.202011521), though no citation for the work is given. Further, the authors appear to conflate intramolecular covalent (i.e., orbital overlap) and non-covalent (i.e., electrostatics, dispersion, etc.) interactions in their descriptor, though the title and basis of the work is focused on “intramolecular noncovalent interactions”; if the authors do not want to conflate these items, a definition of when a bond is formed or not should be provided, especially since the orbital term seems to play a significant role. There are also several concerns regarding the physical understanding provided, given many works in this field that provide clear delineations of these effects.

Our reply: Thanks for the reviewers' comments.

(i) We have cited Angewandte Chemie, 2020, DOI: 10.1002/anie.202011521 for **ref. 57** in the revised manuscript. In the reference, the quantitative relationship between dihedral angle and structural disorder was constructed by using a statistical way of quantifying the planarity of a wide range of conjugated systems, which is similar to the contribution of θ in our descriptor $S = (-\cos \alpha) \cdot \cos^2 \theta \cdot (1 - e^{\Delta \epsilon})^2$. The noncovalent interactions cannot be well characterized only by the dihedral angle.

(ii) The method of NBO analysis is introduced in supplementary information: The NBO perturbative framework applies qualitative concepts of valence theory to describe the noncovalent energy lowering [Chem. Rev. **88**, 899-926 (1988)]. NBO protocols partition electron density from diffuse molecular orbitals into localized Lewis-type orbitals, from which the energy of mixing can be computed. As shown in **Chart S1**, it depicts the interaction of a filled orbital n of the formal Lewis structure with one of the unfilled

antibonding orbitals σ^* to give the second-order energy lowering ($E^{(2)}$). This energy lowering is given by the formula:

$$E^{(2)} = \Delta E_{n\sigma^*}^{(2)} = -2 \frac{\langle n | \mathbf{F} | \sigma^* \rangle^2}{\epsilon_{\sigma^*} - \epsilon_n} \quad (1)$$

Chart S1. Perturbative donor-acceptor interaction, involving a filled orbital n and an unfilled orbital σ^* . \mathbf{F} is the Fock operator and ϵ_n and ϵ_{σ^*} are NBO orbital energies.

The NBO analysis and eq. (1) are general, which can be used to describe the covalent bond and intermolecular or intramolecular noncovalent interactions. When calculating the orbital interaction, it should first define the studied atoms and orbitals which come from the bonding or non-bonding atoms. The method has been widely used to evaluate the strength of intramolecular noncovalent interactions in many works [*Nat. Commun.* **12**, 442 (2021), *J. Am. Chem. Soc.* **140**, 17606-17611 (2018), *Acc. Chem. Res.* **50**, 1838-1846 (2017)]. Here, we apply the method to compute the orbital interaction between **two atoms that are not bonded**, namely $X \cdots Y$ in $\mathbf{M}(X \cdots Y)$ and $\mathbf{PhM}(X \cdots Y)$ molecules and $H \cdots Y$ in $\mathbf{PhM-H}(X \cdots Y)$, and the resultant values are less than 5.27 kcal/mol which are much smaller than the typical value of the covalent C-C bond (ca. 15.86 kcal/mol of C9-C10 bond in the following figure)

There is still a controversy on whether the nature of intramolecular noncovalent interaction like $X \cdots Y$ in organic systems originates from orbital interaction or electrostatic

interaction. Note that many works have been done to clarify the contributions of orbital interaction and electrostatic interactions in different systems [*J. Am. Chem. Soc.* **139**, 15160-15167 (2017), *Angew. Chem. Int. Ed.* **59**, 14602-14608 (2020), *Acc. Chem. Res.* **53**, 2705-2714 (2020)]. In this work, it is found that orbital interactions dominate the intramolecular noncovalent interactions of X...Y in the studied systems.

(iii) As the reviewer said that there are really many works in this field that provide clear delineations of these effects. For example, the noncovalent interactions have been quantitatively calculated to explain the optoelectronic properties of organic semiconductors, and the noncovalent interactions enhance the planarity and rigidity of organic semiconductors (these can be found in introduction section in the main text). However, a direct quantitative relationship between molecular geometry, strength of noncovalent interaction and optoelectronic property is lacking, which is very important for molecular designs and property predictions. In our work, based on the general understandings, a quantitative descriptor S consisting of several geometrical structure parameters is proposed to exhibit excellent positive correlation with noncovalent interaction energy of X...Y NoCLs in organic semiconductors, supported by experimental results, which is very important for rational molecular design and large-scale molecular screening of high-performance organic/polymeric semiconducting materials.

RE: *Recent work from Karunasena and co-workers (Chem. Mater., 2021, DOI: 10.1021/acs.chemmater.1c02335) suggests rather that pi delocalization across the carbon backbone is more important than intramolecular non-covalent interactions in determining conformations. It would be interesting for the authors to evaluate this aspect. In addition to this work, Kharandiuk and co-workers (Chem. Mater., 2019, DOI: 10.1021/acs.chemmater.9b01886) also use atoms-in-molecules (AIM) theory to examine similar effects discussed in this paper.*

Our reply: We really appreciate the reviewers' comments.

(i) We agree with the reviewers that π delocalization across the carbon backbone is more important than intramolecular noncovalent interactions in determining conformations. Therefore, the same carbon backbones are chosen to study the intramolecular noncovalent

conformational locks and their effects on the optoelectronic property in organic semiconductors in our works, which can be seen in **scheme 1**.

(ii) As the reviewer advised, QTAIM is a similar analysis method to routinely explore the topological features of intramolecular noncovalent interaction, where the noncovalent interactions are characterized through electron density (ρ) at the bond critical points (BCPs). From the following **Figure**, it is observed that the intensities of NoCLs (ρ values) vary in the order of $S\cdots O < Se\cdots O < Te\cdots O$, $S\cdots F < Se\cdots F < Te\cdots F$ and $X\cdots F < X\cdots O$ interactions by QTAIM, which is in good agreement with that obtained by the $E^{(2)}$. However, we think the QTAIM calculations are not related to the theme of our work, thus they are not added in the revised manuscript.

Figure. The bond critical points (BCP) with the values of electron density ρ ($e/\text{\AA}^3$) of $X\cdots F$ and $X\cdots O$ interactions by the QTAIM method for the studied systems in the gas phase.

RE: From a computational perspective, sufficient information is provided to generally

repeat the calculations. However, there are several questions pertaining to the methods deployed. First, B3LYP is well established to over-delocalize wavefunctions, which will impact the orbital analyses carried out (especially when the identifier is about 0.1 Ang). Further, B3LYP tends to favor CT-like interactions, which may also influence the descriptions. Hence, it is not clear how dependent the analyses are on a functional that suffers greatly from multi-electron self-interaction errors (MESIE). It is suggested that the authors make use of range-separated hybrid functionals to correct for MESIE. (Note that the CCSD(T) results are probably also influenced by this error given that the geometry used comes from the DFT calculations.).

Our reply: We really appreciate the reviewers' comments and suggestions. Taking the reviewers' advice, we have made use of range-separated hybrid function (ω B97XD) to re-optimize the geometries of the 11 compounds with strong NoCLs and recalculate the corresponding descriptors with results added in **Figures 3d** and **S6** in the main text. Moreover, the CCSD(T) calculations have been performed based on the newly optimized geometries by ω B97XD as shown in **Figure 3d**. Compared the results in **Figure 3c** and **3d**, the ω B97XD/def2-TZVP afforded better correlation with larger linear fitting coefficient ($R^2=0.984$) than B3LYP(D3) of $R^2=0.974$, while the CCSD(T)/def2-TZVP based on the geometries by ω B97XD/def2-TZVP gave the best linear correlation. These results proved that the descriptor is very reliable.

Figure 3. The $E^{(2)}$ versus S baat different theory levels based on the optimized geometries by B3LYP(D3)/def2-TZVP (c) and ω B97XD/def2-TZVP (d) for 11 compounds with apparent noncovalent interactions.

Figure S6. The comparison of the descriptor S for 11 compounds obtained by B3LYP(D3)/def2-TZVP and ω B97XD/def2-TZVP.

RE: All dihedral scans should be carried out from 0 to 180 degrees. For the systems that only went to 90 degrees, a secondary minimum should also be present.

Our reply: Thanks for the reviewers' comments. We have added the PES of Type I from 0 to 180 degrees in **Figure 1** in the revised text. The secondary minimum point appears at $\theta = 130^\circ$, which is an extremely unstable point because of much higher potential energy than the main minimum and small energy barrier of 0.51 kcal/mol (less than the available thermal energy ca. 0.6 kcal/mol at 298 K). It should be noted that the alkyl chains are shorten from $-C_5H_{11}$ to $-CH_3$ because the reasonable optimization geometries cannot be obtained at $\theta = 90^\circ \sim 110^\circ$ owing to too small space for two alkyl chains for $M(S\cdots O)$.

Figure 1. The evolution of (a, c) the atomic distance, (b, d) the potential energy E , the variation of orbital energy $E^{(2)}$ and electrostatic interaction energy E_{elec} as a function of

the θ for $\text{M}(\text{S}\cdots\text{O})$ and $\text{PhM}(\text{S}\cdots\text{O})$.

RE: kT at room temperature is 0.6 kcal/mol. It is difficult to say that one well is much more stable than the other when comparing the minima. Rather, what is important is the barrier to rotation, as the intramolecular interactions tend to be very weak.

Our reply: We really appreciate the reviewer's comments. We have added the torsional PES at B3LYP(D3)/6-31+G(d) level of Type II, as shown in **Figure S2** in SI. Strength of $\sim 2kT$ (ca. 1.2 kcal/mol) is generally considered as the lowest limit for conformational transformation, and the 1.2 kcal/mol is provided as a reference line (dashed line) [*J Phys Chem Lett.* **7**, 3609-3615 (2016)]. From **Figure S2**, it can be known that two conformations (**PhM** and **PhM-H**) coexist, namely $\text{PhM}(\text{S}\cdots\text{C})$ and $\text{PhM-H}(\text{S}\cdots\text{C})$, $\text{PhM}(\text{Se}\cdots\text{C})$ and $\text{PhM-H}(\text{Se}\cdots\text{C})$ and $\text{PhM}(\text{Te}\cdots\text{C})$ and $\text{PhM-H}(\text{Te}\cdots\text{C})$ owing to low energy barrier of <1.2 kcal/mol between them. There is only one conformation for the others because of large energy barrier.

Figure S2. Torsional (dihedral) potential energy surfaces (PES) derived at the

B3LYP(D3)/6-31+G (d) level of type II.

RE: *The authors' discussion on S-O interactions in PhM and PhM-H appears to be number-crunching. The authors should provide a rationale for why orbital interactions are strong.*

Our reply: We really appreciate the reviewer's comments. The orbital interaction has been widely used to investigate the noncovalent interaction in chemistry and biology [*J. Am. Chem. Soc.* **140**, 17606-17611 (2018), *Acc. Chem. Res.* **50**, 1838-1846 (2017), *Nat Commun.* **11**, 2921 (2020), *Inorg Chem.* **59**, 17811-17821 (2020), *J Phys Chem A.* **123**, 5995-6002 (2019), *J. Med. Chem.* **58**, 4383-4438 (2015)]. In S...O interaction, lone pair of oxygen atom resides in sp²-type orbital, which could interact with nearby the σ* orbital of the S-C bond. In particular, donation of lone-pair electron density into the σ* orbital of an adjacent S-C bond could lead to a strong n→σ* interaction.

RE: *"Strength of NoCL exhibits the order of S...F < Se...F < Te...F, S...O < Se...O < Te...O, and X...F < X...O (X=S, Se and Te)". An explanation for the observed trend would be beneficial.*

Our reply: Thanks for the reviewer's suggestion. The atomic radius, and orbital shape and orientation, the feature of orbital interaction have been further analyzed for the order of the strength of NoCLs in the revised manuscript. Taking M(X...F) as an example, from S to Te, the orbital type of interaction, molecular planarity and distance of X...F are all almost unchanged, only the atomic radius gradually increases which results in the enhancement of Δd (Δd = dv - d, dv is the sum of van der Waals radii of two atoms (dv) minus the distance between the two atoms (d)). So the NoCLs is strengthened in the order from S to Se according to $S = (-\cos\alpha) \cdot \cos^2\theta \cdot (1 - e^{\Delta d})^2$. It is easily understood that the increase of atomic radius is naturally beneficial to the overlap between orbitals. The related discussions have been added in the revised manuscript.

RE: *Page 9, line 177. The molecules in this investigation should have sp³ and sp² hybrid orbitals for an organic chemist. The authors should clarify the mention of 'hybrid sp-*

orbital'.

Our reply: We really appreciate the reviewer's comments. We have changed sp to sp² hybrid orbitals in the main text.

RE: While the experiments confirm theoretical findings, the dataset is small to generalize the findings. The authors should expand the dataset by generating the S values and E^2 for all structures used in estimating the PES. This should be more conclusive that the S descriptor performs well for all geometries, not just equilibrium geometry.

Our reply: Taking the reviewer's suggestion, we added the correlation between $E^{(2)}$ and S for 56 compounds in **Figure S7**, and there still is a good linear correlation with $R^2=0.907$.

Figure S7. The correlation between $E^{(2)}$ and S for 56 compounds at B3LYP(D3)/6-31G+(d) level.

RE: The discussion of the reorganization energy is confusing. The molecules are not expected to undergo significant changes in twisting as they are already close to planar. Analyses of the vibrational modes (Huang-Rhys factors, etc) will show that most of the reorganization is due to the changes in carbon-carbon bond lengths, as is well established for π -conjugated systems. So, it is not clear that that the correlation says anything about the systems under investigation.

Our reply: Thanks a lot for the reviewer's comments. We agree with the reviewers. For π -conjugated completely planar and rigid systems, the introduction of NoCLs would not have any effect on the change of conformation upon photoexcitation or electrical

excitation (if so, it is not meaning to introduce the NoCLs). In this case, the reorganization energy mainly comes from the carbon-carbon stretching vibrations and would not have any change before and after NoCLs. Thus, the relationship between reorganization energy and descriptor S no longer exists. That is, if the introduction of NoCLs does not change molecular conformation, the relationship between the strength of NoCLs and reorganization energy would be not exist. Generally, the NoCLs always more and less increase the rigidity of the molecule, and then the relationship would exist whether the effect is obvious or not.

RE: *The comparison of the TDDFT results with the experimental UV/vis shows that the optimized geometries are good estimates of conformations available in solution. There really is no comparison to be made to the proposed descriptor.*

Our reply: Taking the reviewers' suggestion, we have added the comparison of the change of calculated reorganization energy and experimental Stokes shifts versus the descriptor S from **PhM(X··Y)** with NoCLs to **PhM(X··C)** without NoCLs. It is obvious that the correlation between experimental $\Delta\kappa$ and S is very good.

Figure S8. The correlation between $\Delta\lambda_{\text{opt}}$ and experimental $\Delta\kappa$ and S for the compounds with noncovalent interactions. $\Delta\lambda_{\text{opt}}$ and $\Delta\kappa$ are the differences of the reorganization energy and Stokes shifts between **PhM(X··Y)** and **PhM(X··C)**, respectively.

RE: *Why is the analysis of descriptor S for optical reorganization energy restricted to only PhM systems and PhM-H(O--O)? The authors should perform the analysis for all*

56 molecules for completeness.

Our reply: We really appreciate the reviewer's comments. It is always difficult to obtain the equilibrium geometries in the first excited singlet state for $\mathbf{M}(X\cdots Y)$ by using B3LYP(D3) or ω B97XD or other functional. The optimized geometries are always weird and unreasonable as shown in the following **Figure**. So $\mathbf{M}(X\cdots Y)$ systems were not chosen for the analysis of S for optical organization energy. **PhM**($X\cdots Y$) exhibit strong NoCLs, and **PhM-H**($X\cdots Y$) systems mostly behave weak NoCLs. This is convenient to study the role of NoCLs. Therefore, the analysis of descriptor S for optical reorganization energy restricted to **PhM** and **PhM-H** systems.

Figure. The geometric structures of S_1 state are optimized at B3LYP(D3)/6-31+G(d) level.

RE: The S descriptor seems reasonable for E_2 value estimation. However, the descriptor does not predict the trend in the values for estimating the reorganization energy. For instance, in Figure 4b, the S values increase from 0.5 to 1.5, but the reorganization energy is still close to 0.3 eV. Authors should provide more evidence/data to show the actual applicability of the descriptor, which now appears bleak.

Our reply: We really appreciate the reviewer's comments. As discussed above, the relationship between the reorganization energy and the descriptor S works only in this case that the NoCLs have a certain contribution to molecular conformation. From **Figure 4b**, the S values increase from 0.5 to 1.5, and the reorganization energies increase from 0.30 eV to 1.25 eV. The S behaves a crude negative relationship with the reorganization energy. Furthermore, we find a better applicability of the descriptor that there is a very good linear correlation between the $\Delta\lambda_{\text{opt}}$ or $\Delta\kappa$ and S (added in **Figure S8** of the revised

SI). $\Delta\lambda_{\text{opt}}$ and $\Delta\kappa$ are the differences of the reorganization energy and Stokes shifts between **PhM**(X...Y) with strong NoCLs and **PhM**(X...C) without NoCLs, respectively.

Figure 4b. The S versus the excited reorganization energy of **PhM**(X...O) and **PhM**(X...C).

Figure S8. The correlation between $\Delta\lambda_{\text{opt}}$ and experimental $\Delta\kappa$ and S for the compounds with noncovalent interactions. $\Delta\lambda_{\text{opt}}$ and $\Delta\kappa$ are the differences of the reorganization energy and Stokes shifts between **PhM**(X...Y) and **PhM**(X...C), respectively.

RE: In figure 1f, what do the two green traces represent? There is only one Eelec legend. Our reply: Thank you for the reviewers' comment. We have added the classification clearer in the caption of **Figure 1** in the revision version. In **Figure 1f** of the original manuscript, there are two green lines, one represents E_{elec} of S...O interactions at $\theta \in [0^\circ, 180^\circ]$, the other one is E_{elec} of H...O interactions at $\theta \in [0^\circ, 180^\circ]$. In the revised manuscript, there is one green line which represents E_{elec} of S...O interactions at $\theta \in [0^\circ, 90^\circ]$, and H...O interactions at $\theta \in [90^\circ, 180^\circ]$. The E_{elec} of S...O interactions at $\theta \in [90^\circ, 180^\circ]$, and H...O interactions at $\theta \in [0^\circ, 90^\circ]$ are almost zero, so they are deleted.

RE: *Figure 5 atom color coding for 3D molecular representation should be included in the figure caption to improve readability.*

Our reply: We really appreciate the reviewers' advice. Different atoms have been categorized and labeled in the revised version.

RE: *Figure 5: The authors do not show the crystal structure. Should rephrase it along the lines of "molecular geometry from the single crystal structure."*

Our reply: Thanks so much for the reviewers' advice. We have reorganized the description in the revised version.

RE: *F in equation 1 of Chart S1 in SI is not defined.*

Our reply: Thank you for your suggestions. We have added the definition that \hat{F} is the Fock operator, and ϵ_n and ϵ_{σ^*} are the energies of NBO n-orbital and σ^* -orbital, respectively in SI.

RE: *Abstract line 43: typo 'predication'.*

Our reply: Thanks for your instruction. We have corrected the typo in the revised version.

RE: *Typo 'van der Waals radium' throughout the manuscript and SI.*

Our reply: Thanks for the reviewer's comment. We have corrected the typo as "van der Waals radii" in the revised manuscript.

RE: *Page 12, line 225 'As it well known' something is missing in the phrase.*

Our reply: Thanks for your instruction. We have changed to be 'As it is well known' in the revised version.

RE: *Page 13, line 241: Typo "nonradaitive".*

Our reply: We have corrected to be "nonradiative" in the revised version.

Reviewers' Comments:

Reviewer #1:

Remarks to the Author:

I read the revised manuscript and the responses to the reviews. All the questions I am concerned about are answered correctly. Basically, the comments of other reviewers were reasonable replies and revisions. I recommend the manuscript to be accepted in its present form.

Reviewer #2:

Remarks to the Author:

The authors have adequately replied to my comments and modified manuscript and SI accordingly. As already stated in my previous review this is a high quality study with potential for strong impact and I recommend its publication in Nature Communications.

Reviewer #4:

Remarks to the Author:

The authors have addressed all the concerns, and the revised version of the manuscript is suitable for publication.

REVIEWER COMMENTS

Reviewer #1 (Remarks to the Author):

I read the revised manuscript and the responses to the reviews. All the questions I am concerned about are answered correctly. Basically, the comments of other reviewers were reasonable replies and revisions. I recommend the manuscript to be accepted in its present form.

Reviewer #2 (Remarks to the Author):

The authors have adequately replied to my comments and modified manuscript and SI accordingly. As already stated in my previous review this is a high quality study with potential for strong impact and I recommend its publication in Nature Communications.

Reviewer #4 (Remarks to the Author):

The authors have addressed all the concerns, and the revised version of the manuscript is suitable for publication.

Our reply: We appreciate all reviewer's suggestions and it helps us to improve our paper.